# Simulation of Solvent Extraction Circuits for the Separation of Rare Earth Elements

Keven Turgeon [1,*], Jean-François Boulanger [2] and Claude Bazin [1]

1  Department of Mining, Metallurgy and Materials Engineering, Université Laval, Québec, QC G1V 0A6, Canada; claude.bazin@gmn.ulaval.ca
2  Institut de Recherche en Mines et Environnement, Université du Québec en Abitibi-Témiscaminque, Rouyn-Noranda, QC J9X 5E4, Canada; jean-francois.boulanger@uqat.ca
*  Correspondence: keven.turgeon.1@ulaval.ca

**Abstract:** The separation of Rare Earth Elements (REEs) is an important step in the valorization of REE ore and aims at producing individual rare earth compounds for the market. The separation is carried out industrially by solvent extraction (SX) using interconnected circuits consisting of cascades of mixer-settlers. The design of a REE separation circuit implies the selection of the operating conditions and of the number of mixer-settlers required to achieve a target degree of purity for the separated elements. This design work is either carried out by piloting a circuit or using a mathematical simulation. Independent of the method, the world expertise in this area is limited. This paper describes a simulation method requiring a minimum of calibration effort, which can be used to design a complete REE separation plant. The simulation enables assessment of the effect of the number of mixer-settlers per extraction, the scrubbing and stripping stage, as well as the pH of the aqueous solution and organic-phase contents of free and loaded extractant on the purity of the separated REEs. The simulation tool presented here has been developed from a fundamental analysis of the chemical reactions involved in the solvent extraction process. Unlike most of the simulation methods documented in the literature, the method requires no empirical calibration. The proposed method is validated using data from laboratory batch tests and with published data from continuous pilot and industrial REE separation circuits. The application of the simulation tool is illustrated with the planning of the test conditions for a forthcoming pilot test work and with the simulation of a 9-REE product SX separation plant.

**Keywords:** mathematical model; hydrometallurgy; mixer-settler; equilibrium constants

## 1. Introduction

REEs stand for a family of 15 elements ranging from La to Lu, including Y (although some authors include Sc, it is omitted here as it is rarely associated with the other REEs). The REEs are conveniently divided into three groups [1]:

(1)  The Light REE group or LREE consisting of La, Ce, Pr, and Nd.
(2)  The Medium REE group or MREE or SEG for Samarium, Europium, and Gadolinium.
(3)  The Heavy REE group or HREE for Terbium to Lutetium including Yttrium.

These elements find applications in the making of magnets used for wind turbines and electric motors as well as some types of catalysts and batteries [1,2]. Following China's decision to reduce REE exportation leading to the REE price surge in 2010 [3], several countries recognized REEs as critical materials and began to look at options to secure the supply of these elements by the exploitation of their own resources [4].

Most of the REE production comes from mining ore bodies with an increasing proportion from recycling [5]. In the ore, the 15 REEs are clustered in minerals such as bastnasite and monazite [1]. The separation and precipitation of the REEs into individual rare earths are required as part of any REE ore processing or recycling process. The separation of

REEs is industrially carried out by Solvent extraction (SX) in mixer-settlers. Except for the hydrometallurgical processing of oxidized copper ore [6], uranium ore [7], or cobalt-nickel ores, SX is not commonly used in mineral processing. However, although the principles of SX separation for copper or uranium are similar to those of REEs, the processing of REEs is actually very different from the processing of Cu or U. Indeed, the SX processing for Cu and U is commonly carried out using less than 10 reactors or mixer-settlers, while the separation of REEs may require more than 1000 mixer-settlers [8] because of the chemical similarities between the numerous elements to be separated that impose repeating several times each separation step to achieve the desired product purity. The design of a process that consists of more than 1000 connected reactors poses a significant practical challenge. Indeed, if the plant flowsheet is to be developed through piloting (small-scale implementation of the process), the challenge lies in the piloting itself since operating a circuit consisting of 40 mixer-settlers may necessitate 14 days to stabilize [9]. One can thus anticipate that the effort and financial investment for the piloting of a circuit consisting of 1000 units may rapidly discourage motivated REE orebody promoters. This difficulty to pilot REE-SX separation circuits justifies the use of mathematical simulation to design a separation plant [10,11]. The general chemical engineering mathematical approach to conceive a SX plant is based on the McCabe–Thiele diagram [10], an approach that is not adapted to multiple element extraction as is the case of REE separation [11]. The first publications to have proposed a mathematical method to simulate REE separation date back to the 1980s [12,13]. Most of the documented simulation methods use separation factors [11,13,14] as the model parameters for the simulation and, in most cases, the documented applications are limited to one unitary separation (e.g., LREE/HREE, La/Ce, etc.) circuit while a REE separation plant uses several interconnected unitary separation circuits. Indeed, few, if any, papers have attempted to simulate two consecutive separation circuits, say LREEs from HREEs followed by the separation of La + Ce from Pr + Nd, and fewer papers even push further the simulation to individual elements. Even recent publications [11] do not encompass the separation further than one separation stage, confirming the difficulty of extending the applicability of the simulation method to a complete REE separation circuit because of the semi-empirical approach used in these modelling methods.

This paper shows that by relying on the fundamental description of the transfer reactions involved in the SX process and on the application of basic mass conservation equations, it is possible to develop a method to simulate a complete REE-SX separation plant with limited experimental effort. The method explicitly accounts for the pH of the aqueous solution and for the concentration of free extractant in the organic phase. The method provides a flexible and robust approach to simulate the extraction, scrubbing, and stripping stages with a given set of model parameters that can readily be estimated from a limited number of batch extraction tests [15].

The paper is divided into four sections. The first section reviews the REE ore processing to position the separation process and gives some details on the flowsheet of a separation plant. The second section describes the approach used to simulate the separation process. The third section presents a validation of the proposed simulation method by confronting simulation results to data from laboratory tests and industrial and pilot plant operations. The last section illustrates the application of the method for the planning of a pilot test work and for the simulation of a SX separation plant that can separate a REE concentrate feed into nine individual and bulk REE products.

## 2. REE Separation in the Sequence of Processing REE Ores

The separation of REEs is the last step of the valorization process of a REE ore as shown in Figure 1. All 15 (including Y but not Sc.) REEs are usually clustered in particular minerals such as monazite, bastnasite, or eudialyte [1] and they remain clustered during the mining and ore concentration processing steps that aim at producing a concentrate of REE minerals. The REEs are «de-clustered» at the hydrometallurgical step, which is aimed at:

(1) Leaching (sometimes termed «cracking») the minerals to chemically liberate the REEs from each other into an aqueous solution.

(2) Rejecting impurities such as thorium, aluminum, and iron to produce a purified bulk REE solution from which the REEs can be precipitated as a concentrate of mixed REE oxides, chlorides, or carbonates.

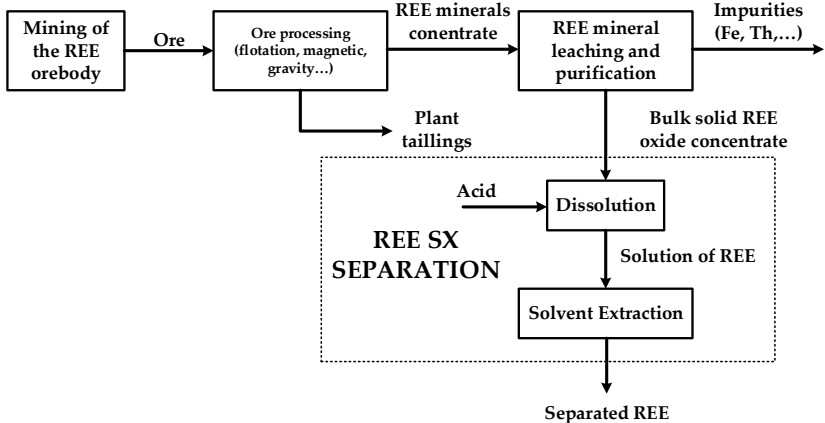

**Figure 1.** Processing steps in the production of individual and/or combined REE products.

The purpose of the separation step is to separate the elements and obtain individual REE oxides or sometimes bulk compounds of two or more REE products that can be sold to the market for further separation or used as is such as the Pr + Nd bulk oxide [16].

Most National Instrument 43-101 studies (NI-43-101) prepared by potential REE Canadian producers in the last decade [17–21] indicate that the final product of their planned project is a mixed or bulk REE oxide concentrate (see Figure 1) to be sold to Chinese refineries that will undertake the separation. However, refiners often impose important refining charges and have the flexibility of being very demanding on the composition of the concentrate. These constraints significantly reduce the profitability of REE projects and maintain China's control of the supply chain. A few potential Canadian REE producers have indeed considered the idea of performing the separation of their produced bulk REE oxide concentrate but finally gave up. Recently, an Australian company has recognized the economic advantage of integrating REE separation into its project [22]. Numerous reasons can explain the hesitation of promotors in considering a separation step for their REE project:

- Separation of REEs is marginally known outside China, except for the now-ceased production of yttrium in Canada [23], intermittent operation at Mountain Pass, USA [24], and the Lynas plant in Malaysia [25], and the details of these non-Chinese operations are seldom available to the public.
- Separation of REEs is a complex process and it is different from known conventional mineral processing or even hydrometallurgical operations.
- There is little public information concerning the industrial aspects of the REE separation.
- There is no detailed public information on capital expenditures and operating costs for REE separation plants.
- There is a very limited number of experts available to help promoters in the design of a REE separation plant.

It rapidly becomes obvious that this lack of information and limited accessibility to the expertise of operating REE separation plants creates hesitation when considering separation as an integrated part of a REE exploitation process. Mathematical modelling and simulation provide an economic platform to generate knowledge about a process [26]. A versatile REE-SX separation plant simulator can indeed be useful for plant design purposes and to assess the consumption of reagents (namely, acid and base) and the number of mixer-settlers and, thus, to help in estimating the initial Operating Expenditures (OpEx) and

Capital Expenditures (CapEx) for a process. The simulation tool can also find applications in process optimization for the selection of the REE products that should be economically released from a SX Separation plant [27,28].

## 2.1. Separation of REEs

The separation of REEs typically starts with the preparation of an aqueous solution by digesting the bulk mixed REE oxide concentrate in an acid (see Figure 2), usually HCl. The Pregnant Leach Solution (PLS) thus contains the 15 lanthanides ($La^{3+}$ to $Lu^{3+}$) and yttrium to be separated into individual compounds. There are various separation strategies [1,14,29]. Figure 2 illustrates a possible strategy. For economic reasons, cerium should be separated (possibly precipitated by oxidation to $Ce^{4+}$) prior to the SX separation plant [30]. Most of the separation processes begin with a separation of the LREEs from the SEG and HREEs (SX-1 in Figure 2) and continue by separating the isolated sub-groups (e.g., LREE) into their constituents (e.g., La-Ce/Pr-Nd) and carry on until a fairly pure (>99%) solution of each element is obtained.

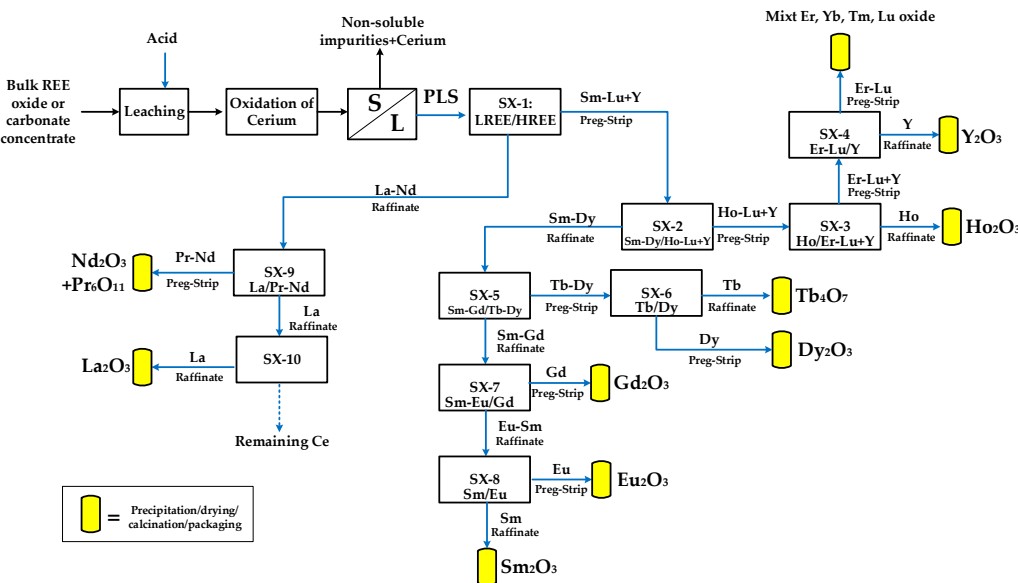

**Figure 2.** Possible REE separation scheme to obtain 9 individual REEs and two bulk REE products. (Note: Case where cerium is mostly removed prior to the SX plant.)

SX is currently the only industrial method used [29] to separate the REEs contained in an aqueous solution. Numerous textbooks [31,32] describe the principles of SX, which consists of mixing the aqueous solution containing the elements to be separated with a non-miscible organic phase that contains an extractant (also called ligand) designed to complex with targeted elements of the aqueous solution. Figure 3 shows a single REE separation circuit that aims at separating elements of group A from the elements of group B. This unitary circuit usually consists of an Extraction step in which mostly B elements are transferred into an organic phase, leaving the A elements in the aqueous phase. The Extraction step is followed by a scrubbing step that aims at washing the organic phase from the co-extracted A elements. The scrubbed organic phase is finally stripped of the B elements in the Stripping step. The depleted organic phase is finally recirculated to the extraction step, after a possible conditioning step (see Figure 3).

Several repetitions of the extraction and scrubbing steps are required to obtain the desired degree of purity of the organic phase. The required number of contacts to achieve a targeted product purity (usually +99%) are parameters that need to be estimated for each separation circuit (SX boxes in Figure 2).

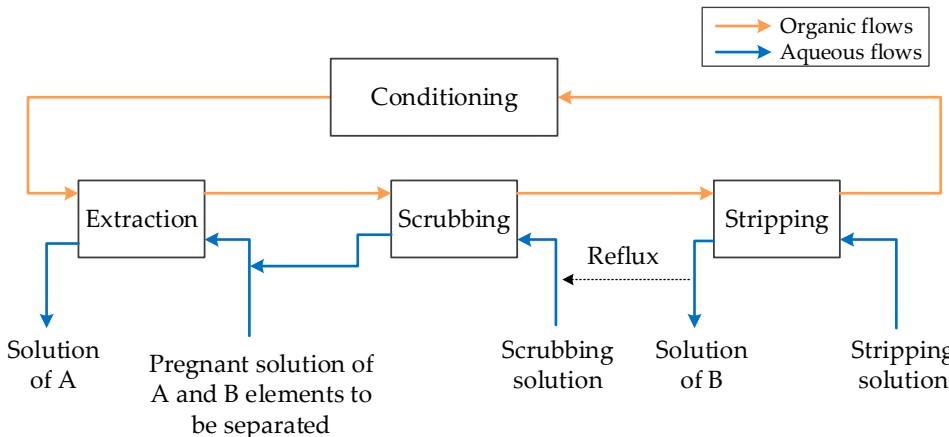

**Figure 3.** Extraction, scrubbing, and stripping steps in the separation of Elements A from Elements B.

The mass transfers that occur during the extraction, scrubbing, and stripping stages are described by the following reactions:

$$M^{n+}_{e;Aq} + nRH_{Org} = R_nM_{e;Org} + nH^+_{Aq} \text{ for } e = 1, \ 2, \ \dots \ N_e \tag{1}$$

$$M^{n+}_{e2;Aq} + R_nM_{e1;Org} = R_nM_{e2;Org} + M^{n+}_{e1;Aq} \tag{2}$$

where *RH* stands for a cationic extractant molecule [14]. The extractant molecule is solubilized in an organic diluent, usually kerosene, forming the organic phase that is immiscible in the aqueous solution. The variable $M_e$ stands for any element (REE or impurity) in the aqueous solution. The subscripts $_{Aq}$ and $_{Org}$ of Equations (1) and (2) indicate the aqueous and organic phases. Equation (1) can also be expressed with the extractant in its dimer form, i.e., $H_2R_2$, but the equations are similar whether in the presence of the monomer or dimer form of the extractant [15]. Equation (2) describes the substitution of a less stable element $M_{e;1}$ by a more stable one $M_{e;2}$. The reaction of Equation (1) describes either the extraction, scrubbing, or stripping processes. Simply applying Le Chatelier's principle shows that the equilibrium of the reaction can be displaced by adjusting the pH of the aqueous solution. The application of this fundamental law makes possible the formulation of the flexible process model described in this paper.

### 2.2. Equipment for the SX Separation of REEs

The operations of mixing the aqueous and organic phases and separating them are readily carried out in mixer-settlers (MS) that are connected together to carry out the operations shown in Figure 3. A simplified schematic of a MS is shown in Figure 4 with a battery of interconnected MS to repeat the loading, scrubbing, or stripping actions as required to obtain a pure product. The agitator of the mixer of a MS mixes the phases but also acts as a pump allowing the transfer of the aqueous and organic phases between the MS without the need for external pumps.

A separation stage as indicated by each SX-block in Figure 2 is thus a process involving the extraction, scrubbing, and stripping steps each consisting of several contacts as illustrated in Figure 4. Despite the apparent complexity of the process in Figure 3, the Conditioning–Extraction–Scrubbing–Stripping (CESS) process yields only two main output aqueous streams, namely the extraction raffinate that contains the A elements and the strip solution that contains the B elements (see Figure 3). This can be seen in Figure 2 where each SX block has effectively two output streams directed toward another separation process or a precipitation process to recover the elements from the solution.

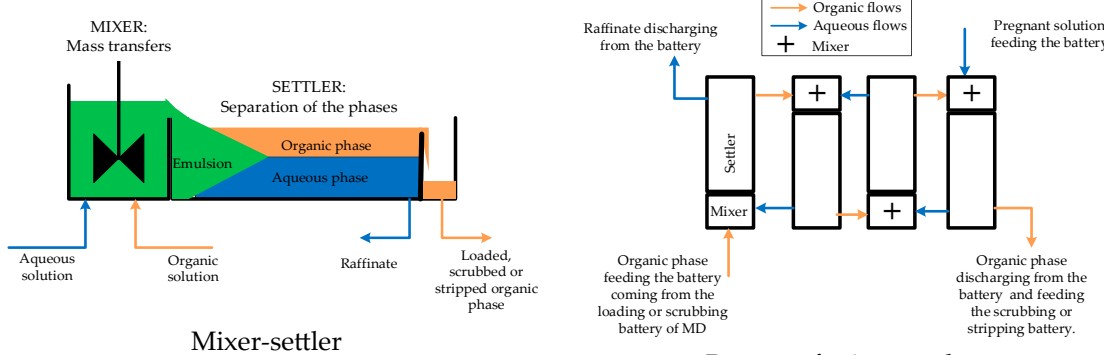

**Figure 4.** Mixer-settler and battery of mixer-settlers.

The conditioning step in Figure 3 refers to either the washing of the organic phase to remove entrained acid from the stripping or saponifying the organic phase to substitute the protons H$^+$ of the extractant by Na$^+$ in order to minimize the acidification of the aqueous phase during the extraction step [14].

When reactions of Equations (1) and (2) are allowed to evolve for a sufficiently long period of time (e.g., 5 to 30 min, depending upon the extractant and mixing hydrodynamics) an equilibrium is reached [11]. Most simulation approaches, as well as the one proposed here, assume that the equilibrium is reached during the mixing of the phases; so, when the mixture of aqueous and organic phases leaves the mixing unit of a MS, all the elements of the two phases are in equilibrium. The mixture then overflows into the settler where the aqueous and organic phases separate due to their immiscibility and difference in specific gravity (see Figure 4). All the REE-SX simulation models assume that as the phases leave the mixer, a chemical equilibrium is established.

## 3. Mathematical Model of the REE-SX Separation

Since the operations of extraction, scrubbing, and stripping are all described by Equations (1) and (2), it is necessary only to find a way to predict the equilibrium conditions from the initial composition of the aqueous and organic phases feeding the mixer to simulate the mass transfers occurring in a mixer-settler (MS). The MS mathematical model is thus the basic brick in the development of a tool to simulate a complete REE separation circuit as the one shown in Figure 2.

### 3.1. Notations, Material Balance, and Model Equations to Simulate a Mixer-Settler

The basis of the proposed model is to assume that the mass transfers of Equations (1) and (2) have reached an equilibrium at the discharge of the mixer. The equilibrium for the reaction:

$$M^{n+}_{e;Aq} + nHR_{Org} = R_nM_{e;Org} + nH^+_{Aq} \text{ for } e = 1, \ 2, \ \dots \ N_e \tag{3}$$

is characterized by the thermodynamic constant [11]:

$$K_{Eq;Me} = \frac{a^{eq}_{R_nM_e;Org}}{a^{eq}_{M_e^{n+};aq}} \times \left( \frac{a_{H^+;Aq}}{a_{HR;Org}} \right)^n \tag{4}$$

where the variable $a^{eq}_{z;P}$ in Equation (4) is the activity of species $z$ in phase $P_{(Aq} $ or $_{Org})$ at the equilibrium ($^{eq}$). Using the assumption that the aqueous and organic phases are ideal solutions or follow Henry's law, which is valid for dilute solutions as most of those in SX, the activity of a species can be approximated by its molar concentration [11]. The ideal solution hypothesis has been verified for the case of the extraction and scrubbing solutions [33], but validation remains to be performed for concentrated solutions, as in

the case of stripping, where a concentrated acid is used to return the elements of a stable complex into an aqueous solution [34].

The objective of the simulation is thus to find the equilibrium concentrations of all the species in solution that verify the constraint of Equation (4). The equilibrium concentrations are deduced from material balance, i.e.,

$$Q_{Aq}x_e^{eq} = Q_{Aq}x_e^0 - \Delta_e \text{ for } e = 1 \text{ to } N_e \tag{5}$$

$$Q_{Org}y_{RMe}^{eq} = Q_{Org}y_{RMe}^0 + \Delta_e \text{ for } e = 1 \text{ to } N_e \tag{6}$$

where $x_e^k$ stands for the molar (mole/L) concentration of element e in the aqueous phase in the input of the mixer ($k = 0$) and at equilibrium or discharge of the mixer ($k = eq$). The variable $y$ stands for a molar concentration in the organic phase. The number of elements (REEs and impurities) that can transfer between the phases is noted $N_e$. The volumetric flow rate (e.g., L/min) of aqueous solution is noted $Q_{Aq}$ and $\Delta_e$ is the rate (mole/min) of moles of element $e$ transferred between the aqueous and organic phases. From Equations (5) and (6), it follows that:

$$x_e^{eq} = \frac{Q_{Aq}x_e^0 - \Delta_e}{Q_{Aq}} \text{ for } e = 1 \text{ to } N_e \tag{7}$$

$$y_{RMe}^{eq} = \frac{Q_{Org}x_{RMe}^0 + \Delta_e}{Q_{Org}} \tag{8}$$

Assuming that element $e$ in solution is under an oxidation state ($x_e^{n_e^+}$), the mass balances for the proton and free extractant become, respectively:

$$Q_{Aq}x_{H+}^{eq} = Q_{Aq}x_{H+}^0 + \sum_{e=1}^{Ne} n_e\Delta_e \tag{9}$$

$$Q_{Org}x_{RH}^{eq} = Q_{Org}x_{RH}^0 - \sum_{e=1}^{Ne} n_e\Delta_e \tag{10}$$

or:

$$x_{H+}^{eq} = \frac{Q_{Aq}x_{H+}^0 + \sum_{e=1}^{Ne} n_e\Delta_e}{Q_{Aq}} \tag{11}$$

$$y_{RH}^{eq} = \frac{Q_{Org}y_{RH}^0 - \sum_{e=1}^{Ne} n_e\Delta_e}{Q_{Org}} \tag{12}$$

Equations (11) and (12) are general and account for elements with different oxidation numbers such as calcium ($Ca^{2+}$) knowing that most of the REEs are trivalent (3+). The sign of $\Delta_e$ is either positive or negative according to the simulated operation (extraction, scrubbing, or stripping). Under the ideal solution hypothesis, the equilibrium concentrations of Equations (7), (8), (11), and (12) allow us to write:

$$K_{Eq;e} = \frac{a_{R_nMe;Org}^{eq}}{a_{M_e^{n+};aq}^{eq}} \times \left(\frac{a_{H+;Aq}}{a_{HR;Org}}\right)^n = \frac{\frac{Q_{Org}y_{RMe}^0 - \Delta_e}{Q_{Org}}}{\frac{Q_{Aq}x_e^0 - \Delta_e}{Q_{Aq}}} \times \left(\frac{\frac{Q_{Aq}x_{H+}^0 + \sum_{e=1}^{Ne} n_e\Delta_e}{Q_{Aq}}}{\frac{Q_{Org}y_{RH}^0 - \sum_{e=1}^{Ne} n_e\Delta_e}{Q_{Org}}}\right)^{n_e}, \tag{13}$$

$$\text{for } e = 1 \text{ to } N_e$$

So, with values for the equilibrium constants for each element (REEs or impurities) transferred between the phases (calibration parameters), $Q_A$ and $Q_{org}$ (operating parameters) and composition of the feed aqueous and organic solutions $x_e^0$, $y_{RMe}^0$, $x_{H+}^0$, and $y_{RH}^0$ (operating parameters), Equation (13) is a system consisting of $N_e$ equations with $N_e$ unknown, the $\Delta_e$ values. The system of equations is thus perfectly defined although strongly non-linear in $\Delta_e$. A robust iterative numerical method was developed to solve the system

of equations [28]. Equation (13) can thus be used to simulate the extraction, scrubbing, or stripping operations. The initial conditions ($x_e^0$, $y_{RMe}^0$, $x_{H+}^0$, and $y_{RH}^0$) define the type of operation occurring in the MS (extraction, scrubbing, or stripping), a feature that is particular to the modelling approach proposed here. Indeed, other simulation approaches require that the user identifies the operation that takes place in a MS [12].

The input information required to simulate the scrubbing, extraction, or stripping stage of a separation circuit is thus simply the operating conditions ($Q_A$ and $Q_{org}$, $x_e^0$, $y_{RMe}^0$, $x_{H+}^0$, and $y_{RH}^0$) and the equilibrium constants ($K_{Eq;e}$). The equilibrium constants are in fact the only calibration parameters that need to be obtained through experimentation and are sufficient to simulate a complete separation circuit as in Figure 2, as long as the same extractant, diluent, and temperature are used throughout the whole circuit.

### 3.2. Model of a Battery of Mixer Settlers

A battery of MS consists of a cascade of *N* mixer-settlers that exchange the discharge of adjacent settlers. The difficulty associated with the simulation of a MS battery is related to the fact that the aqueous solution is flowing counter current to the organic phase, as shown in Figure 5, in order to maximize the concentration gradient between the exchanged species [31,32]. In this counter current scheme, the composition of the organic phase entering the battery is known, and it is contacted with the raffinate of the second MS of the battery whose composition is unknown, so the first MS of the battery cannot be simulated by the above procedure. The same difficulty arises for the Nth MS of the battery that is fed by an aqueous solution of known composition and the organic solution discharged from the N-1th MS of the battery that has an unknown composition. Despite this difficulty, the problem of simulating a battery of MS is perfectly defined with the same number of equations and unknowns.

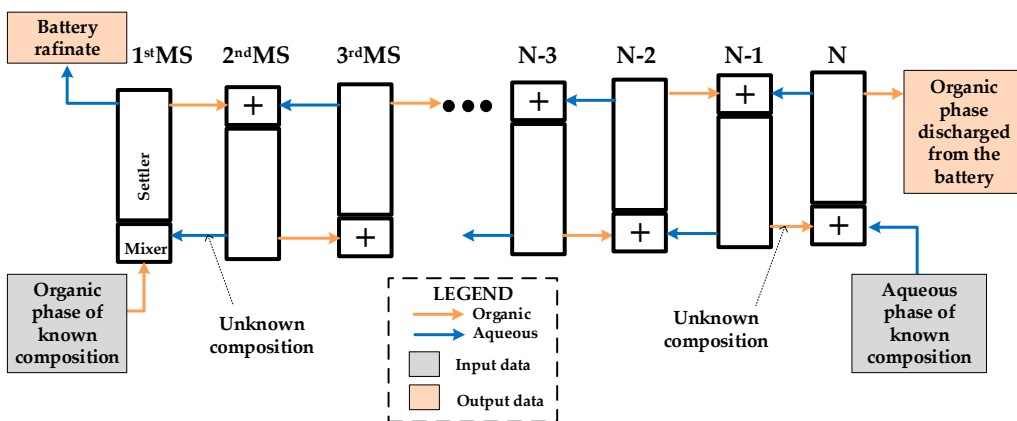

**Figure 5.** Aqueous and organic phases flowing counter-current in a battery of MS.

The solution to the problem used here is iterative. It starts with initial guesses for the composition of the organic phase exchanged between all the MSs of the battery. Then, starting with the composition of the organic phase discharged from the second MS (see Figure 5), the first MS is simulated to yield the composition of the aqueous phase discharged from the first MS. The second MS is simulated using the guessed composition of the organic phase discharged from the third mixer allowing the composition of the aqueous phase discharged from the second MS to be calculated. The process is then repeated for all the MSs of the battery until the composition of the raffinate of the last MS converges from one iteration to the following one.

As indicated above, during the extraction step, unwanted elements also transfer into the organic phase. This is the case for elements A for the A/B separation in Figure 3. These unwanted elements form less stable complexes with the extractant than the targeted elements and can be removed from the organic phase by contacting the loaded organic

phase with a slightly acidic aqueous solution (the scrubbing solution) in order to push the reaction of Equation (1) toward the left, i.e.,

$$X_n A_{Org} + nH^+_{Aq} \rightarrow A^{n+}_{Aq} + nHX_{Org} \tag{14}$$

The *A* elements in the organic phase can also be dislodged by *B* elements present in the scrubbing solution (Equation (2)):

$$X_n A_{Org} + B^{n+}_{Aq} \rightarrow A^{n+}_{Aq} + X_n B_{Org} \tag{15}$$

Reaction (15) plays an important role in the scrubbing, particularly when a portion of the stripped loaded solution is reverted to the scrubbing battery (see Figure 3) as a way to minimize acid consumption [11]. It can be demonstrated that when the equilibrium conditions are found using the above procedure, these equilibrium conditions verify the equilibrium conditions for Equation (15) [28]. Therefore, the proposed modelling approach implicitly accounts for Equation (15) (scrubbing of *A* by *B*), while it is not clear if the other models for REE-SX separation incorporate Equation (15). The raffinate from the scrubbing battery is usually completely reverted to the extraction battery (see Figure 3) in order to recover A and B elements that are in the scrubbing raffinate.

The assumption of ideal solutions implies that the equilibrium constants estimated from standard batch extraction tests are also valid to describe the scrubbing and stripping operations as long as the temperature and the diluent of the organic phase remain the same. This is the idea behind the mathematical model of the solvent extraction process presented in this paper and was verified experimentally [28].

Equation (1) is also used to simulate stripping. Since stripping uses a concentrated acid solution, some authors have questioned the validity of the ideal solution hypothesis [34]. Nevertheless, none of the simulations carried out so far yielded sufficient arguments to reconsider the working hypothesis of the ideal solution. The only uncertainty is related to the fact that the simulation of the stripping of the heaviest REEs (e.g., Tm, Yb, and Lu) from the loaded organic phase is always incomplete, with less than 3% of these elements remaining in the organic phase. This incomplete stripping is hypothetically attributed to the underestimation of the $H^+$ activity by the molar concentration for the strong acid used for stripping [34,35].

### 3.3. Programming the Mathematical Model

Considering that a MS is the basic block of a REE separation circuit, the mathematical model for a MS is programmed as an «object» using a programming language that supports Object Oriented Programming. Therefore, by providing values for the necessary input variables, the MS «object» can predict the equilibrium compositions of the aqueous and organic phases discharging from the settler. These data can be passed to the next MS of a battery of MS to simulate a battery consisting of a cascade of N mixer settlers. The MS object can thus be used to build an «object battery of MS». Indeed, the «battery object» would only need to input the flow and composition of the aqueous and organic solutions entering the battery of MS and the number of mixer settlers of the battery to predict the composition of the output streams of the battery. With the proposed model, the simulation of the extraction, scrubbing, or stripping stages does not require formulating particular hypotheses or conditions to simulate any of these steps, and the same «battery object» can be used to simulate the extraction, scrubbing, and stripping operations. Once the object for a battery of MS is developed, the logical next step is to combine battery objects to have a «CESS object» to simulate a Conditioning–Extraction–Scrubbing–Stripping (CESS) unitary circuit, and these objects can finally be connected to each other to simulate a complete REE separation circuit as in Figure 2. The only input variables that need to be experimentally calibrated to simulate the whole process are the equilibrium constants for the reaction of Equation (1), which are the same for any of the separation circuits (SX boxes) of Figure 2,

providing that the diluent, extractant, and solution temperature remain identical to those used for the calibration experiments.

### 3.4. Equilibrium Constants

The equilibrium constants are the only model parameters that need to be experimentally determined to enable the simulation of a whole REE separation circuit. These parameters are obtained from standard batch extraction tests conducted with separatory funnels [11]. A robust estimation method was developed to estimate the equilibrium constants from these batch extraction tests planned to generate redundant data favorable to the estimation of reproducible values. Table 1 shows the estimated REE equilibrium constants for some common commercial cationic extractants. The relative amplitudes of these values are consistent with the separation factors provided in the literature despite the variability observed for the reported separation factors given in Table 2.

**Table 1.** Estimated equilibrium constants for different extractants in kerosene at 25 °C.

|  | **D2EHPA** | **Cyanex 572** | **Cyanec 272** | **P507** | **P507 (Dimer)** |
|---|---|---|---|---|---|
| La | $1.97 \times 10^{-3}$ | $2.36 \times 10^{-5}$ | $1.28 \times 10^{-6}$ | $1.95 \times 10^{-3}$ | $1.93 \times 10^{-2}$ |
| Ce | $4.22 \times 10^{-3}$ | $1.63 \times 10^{-5}$ | $1.50 \times 10^{-5}$ | $2.86 \times 10^{-3}$ | $3.05 \times 10^{-2}$ |
| Pr | $4.49 \times 10^{-3}$ | $7.58 \times 10^{-5}$ | $5.09 \times 10^{-5}$ | $4.28 \times 10^{-3}$ | $4.57 \times 10^{-2}$ |
| Nd | $4.79 \times 10^{-3}$ | $1.40 \times 10^{-4}$ | $6.59 \times 10^{-5}$ | $5.33 \times 10^{-3}$ | $5.66 \times 10^{-2}$ |
| Sm | $2.32 \times 10^{-2}$ | $2.59 \times 10^{-3}$ | $8.71 \times 10^{-4}$ | $1.81 \times 10^{-2}$ | $1.95 \times 10^{-1}$ |
| Eu | $5.18 \times 10^{-2}$ | $3.98 \times 10^{-3}$ | $1.17 \times 10^{-3}$ | $1.32 \times 10^{-1}$ | $1.40 \times 10^{0}$ |
| Gd | $8.78 \times 10^{-2}$ | $2.36 \times 10^{-3}$ | $1.35 \times 10^{-3}$ | $1.98 \times 10^{-1}$ | $2.10 \times 10^{0}$ |
| Tb | $1.40 \times 10^{-1}$ | $3.44 \times 10^{-2}$ | $4.35 \times 10^{-3}$ | $1.15 \times 10^{0}$ | $1.22 \times 10^{1}$ |
| Dy | $1.99 \times 10^{-1}$ | $1.83 \times 10^{-1}$ | $8.36 \times 10^{-3}$ | $3.24 \times 10^{0}$ | $3.44 \times 10^{1}$ |
| Ho | $2.46 \times 10^{-1}$ | $2.61 \times 10^{-1}$ | $1.86 \times 10^{-2}$ | $6.48 \times 10^{0}$ | $6.88 \times 10^{1}$ |
| Er | $4.18 \times 10^{-1}$ | $1.33 \times 10^{0}$ | $3.60 \times 10^{-2}$ | $1.77 \times 10^{1}$ | $1.88 \times 10^{2}$ |
| Tm | $6.28 \times 10^{-1}$ | $8.30 \times 10^{-2}$ | $9.58 \times 10^{-2}$ | $5.91 \times 10^{1}$ | $6.28 \times 10^{2}$ |
| Yb | $8.15 \times 10^{-1}$ | $7.88 \times 10^{-1}$ | $2.30 \times 10^{-1}$ | $2.10 \times 10^{2}$ | $2.41 \times 10^{2}$ |
| Lu | $8.37 \times 10^{-1}$ | $1.65 \times 10^{-2}$ | $3.03 \times 10^{-1}$ | $3.75 \times 10^{2}$ | $4.29 \times 10^{2}$ |
| Y | $3.57 \times 10^{-2}$ | $3.18 \times 10^{-1}$ | $3.00 \times 10^{-2}$ | $1.24 \times 10^{1}$ | $1.31 \times 10^{2}$ |
| Sc | $7.58 \times 10^{-1}$ | $2.61 \times 10^{-3}$ | $2.61 \times 10^{-3}$ | $7.58 \times 10^{-1}$ | $7.58 \times 10^{-1}$ |

**Table 2.** Discrepancies between separation factors for P507 reported in the literature (the % variation is the relative standard deviation).

| | **Separation Factors from:** | | | | | | |
|---|---|---|---|---|---|---|---|
| **Elements** | **Balint [36]** | **Sato [37]** | **Han [38]** | **Quan [39]** | **Turgeon [33]** | **Average** | **Variation (%)** |
| Ce/La | 6.83 | 1.30 | 13.84 | 4.32 | 1.58 | 5.57 | 92.2 |
| Pr/Ce | 2.03 | 1.09 | 1.29 | 1.86 | 1.50 | 1.55 | 25.1 |
| Nd/Pr | 1.55 | 1.17 | 3.2 | 1.42 | 1.24 | 1.72 | 49.2 |
| Sm/Nd | 10.6 | 2.00 | 8.36 | 12.4 | 3.44 | 7.36 | 61.1 |
| Eu/Sm | 2.34 | 1.96 | 2.34 | 16.4 | - | 5.76 | 123.2 |

The variability in the separation factors is attributed to differences in the experimental conditions used by the authors. Indeed, it was not possible to find a standardized procedure to estimate the separation factors, a situation that is somehow disturbing considering that separation factors are the basis of most of the simulation tools used to design a REE-SX separation circuit. The use of a standardized and documented procedure to obtain the equilibrium constants for the separation of REEs, such as the one previously presented in [15], could only make the predictions of a simulator based on these parameters more robust. In addition, the proposed calibration procedure for the equilibrium constants is designed to produce redundant data that reduce the sensitivity of the estimates to measurement errors, an aspect seldom discussed in papers giving separation factors.

Another advantage of relying on equilibrium constants rather than separation factors is related to the fact that separation factors do not explicitly account for the pH of the solution nor the free extractant concentration in the organic phase, leading to the necessity of conducting experiments at various pH and extractant concentrations to predict the impact of these variables on the separation factors. The separation factors must then be linked to these conditions through empirical models making difficult the proposal of a generalized approach to simulate a REE-SX separation circuit [11]. This effort is not required when using the equilibrium constants for the simulation. The proposed approach thus releases the experimenters from the necessity of developing empirical relationships to correct the separation factors for the pH and free extractant concentration, and allows the use of the same simulation procedure for the extraction, scrubbing, and stripping stages of any unitary separation circuits (SX blocks in Figure 2).

Usually, the extraction tests conducted to estimate the equilibrium constants or separation factors are carried out with synthetic solutions prepared using commercial REE products. Since the equilibrium constants are used to predict the operation of a separation plant that would process a solution obtained from ore processing, extraction tests were also conducted using an actual PLS produced from a Canadian REE ore [15,33]. The estimated equilibrium constants were found to be statistically not different from those obtained using synthetic solutions, as shown in Table 3. This result does not contradict the hypothesis of ideal solutions that is the basis of the proposed simulation approach.

**Table 3.** Equilibrium constants for Cyanex 572 obtained from a synthetic and an actual REE pregnant leach solution.

| | Synthetic REE Solution | | | Actual REE Solution | | | |
|---|---|---|---|---|---|---|---|
| | Number * | Average | Std-Dev. | Number * | Average | Std-Dev. | *t* Test Value |
| La | 18 | $4.60 \times 10^{-5}$ | $5.70 \times 10^{-5}$ | 3 | $4.30 \times 10^{-5}$ | $1.00 \times 10^{-5}$ | 0.13 |
| Pr | 18 | $1.69 \times 10^{-4}$ | $8.80 \times 10^{-5}$ | 3 | $1.68 \times 10^{-4}$ | $1.01 \times 10^{-4}$ | 0.01 |
| Nd | 18 | $2.80 \times 10^{-4}$ | $1.40 \times 10^{-4}$ | 3 | $2.50 \times 10^{-4}$ | $1.40 \times 10^{-4}$ | 0.24 |
| Sm | 18 | $2.55 \times 10^{-3}$ | $1.80 \times 10^{-3}$ | 3 | $2.47 \times 10^{-3}$ | $1.29 \times 10^{-3}$ | 0.07 |

*: Number of replicated experiments.

### 3.5. Input Variables for the Simulation

The input variables for a CESS object are indicated in Figure 6. Except for the equilibrium constants, the input variables are either operating or design variables that the user needs to define to launch a simulation. The operating variables are the extractant concentration in the organic phase, the organic phase flow rate, the pH of the extraction feed solution and of the scrubbing solution, and the flow rate and composition of the PLS and stripping solutions. The user can also specify the amount of «reflux» or back-recirculation of the scrubbing or stripping solutions, as shown in Figure 3. Usually, 100% of the scrubbing raffinate is reverted to the extraction battery. Except in [11], few simulation tools described in the literature offer this possibility, while reflux provides a way to save on the scrubbing acid solution as the refluxed strip solution is a strong acid and carries elements that can displace (scrub) the impurities out of the organic phase via the reaction of Equation (2). The CESS simulation object is also equipped with pH control loop modules for the solutions feeding the extraction battery and the solution feeding the scrubbing battery (see Figure 6). These control loops are necessary to maintain a target pH in the feed of extraction and scrubbing stages despite the reflux streams. The design variables are the numbers of MS in the extraction, scrubbing, and stripping batteries, the pH set-points for the feed solution to the extraction and scrubbing batteries, and the acid strength of the stripping solution. The discharged aqueous solution from the conditioning battery is either discarded or recycled. In the simulation tool, this aspect is not yet considered. Neglecting the conditioning raffinate and if the scrubbing raffinate is completely recycled to the extraction battery, there are

only two aqueous output streams, namely, the extraction raffinate and strip solution, for which the simulation provides the flowrate and composition.

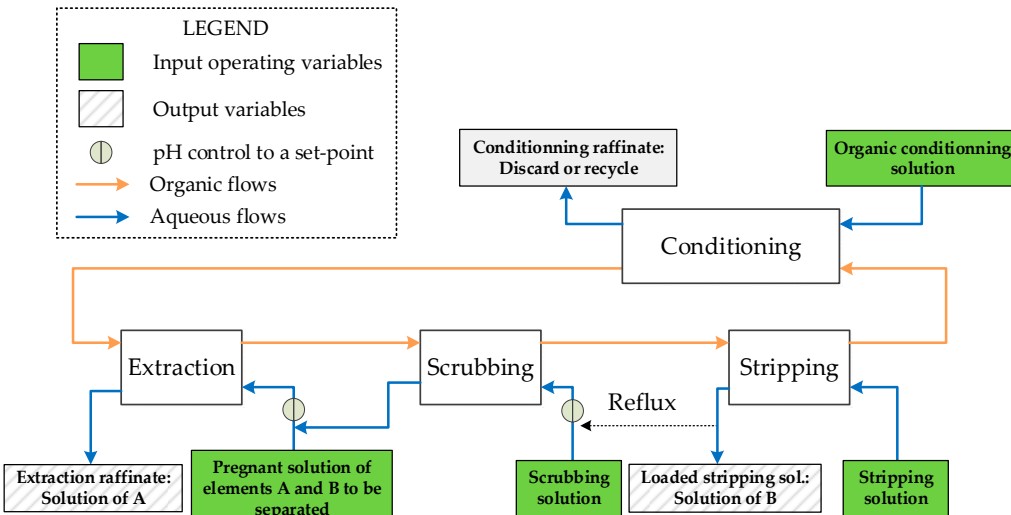

**Figure 6.** Input and output variables for and from the simulation of a CESS-SX separation circuit.

## 4. Validation of the Simulation Method

Three examples are presented in this section to validate the proposed simulation method. The first one uses laboratory tests, while the second and third examples compare actual pilot and industrial plant observed results to simulated ones.

### 4.1. Validation Using Laboratory Tests

The usual practice to validate a model consists in comparing the model predictions to the data used to calibrate the model or *goodness of fit validation*. The approach followed here uses a set of experimental data to estimate the equilibrium constants, and the estimated equilibrium constants are used to simulate extraction experiments conducted independently from the calibration experiments or *good prediction validation*. The good prediction validation is used here, as it measures the robustness of the simulation. Only the extractant, diluent, and temperature of the solution are kept constant between the calibration and validation experiments.

#### 4.1.1. Experimental Methodology

The aqueous PLS were prepared using deionized water and by dissolving rare earth oxides (Thermo Fisher Scientific, Waltham, MA, USA) with HCl (Thermo Fisher Scientific). The organic phase is made of Cyanex 572 extractant (Solvay, Brussel, Belgium) diluted in Elixore 230 (Total, Paris, France). For the extraction tests, about 50 mL of aqueous solution is mixed with the organic phase at a specific organic-to-aqueous volume ratio in separatory funnel of 250 mL, then agitated mechanically for 10 min on a VWR rocking shaker. The REE concentrations of the aqueous solutions are measured using a microwave plasma atomic emission spectrometer (MP-AES 4200, Agilent Technologies, Santa Clara, CA, USA) calibrated with multi-element REE solutions. The pH of the aqueous solutions is measured using an accumet XL600 pH-meter with an Orion ROSS pH probe calibrated from pH 1 to 7 (Thermo Fisher Scientific).

#### 4.1.2. Experimental Tests

Three PLS with the compositions given in Table 4 were prepared and used for the validation. The initial pHs of the solutions range from 1.42 to 2.3. The organic phase is made of a 10% *v/v* solution of extractant Cyanex 572 in Elixore 230. The Cyanex 572 REE equilibrium constants were estimated from previous tests [33] and are given in Table 1. These data are used to simulate the extractions with the four Volumes of organic phase/Volume of aqueous phase ratios (O/A) ratios indicated in Table 4. For a single batch

extraction, such a simulation is readily carried out using Microsoft Excel$^{TM}$. Following the experiments, the concentrations of REEs and the equilibrium pH of the raffinate are measured. The simulated and observed REE extraction ratios are calculated using:

$$R_e = 100\left(1 - \frac{x^{eq}_{e;Aq}}{x^{0}_{e;Aq}}\right) \tag{16}$$

where $R_e$ is the proportion of element e that is transferred from the aqueous to the organic phase, while $x^{0}_{e;Aq}$ and $x^{eq}_{e;Aq}$ are the measured and simulated concentrations of element $e$ in the PLS (superscript $^0$) and in the raffinate or at equilibrium (superscript $^{eq}$).

**Table 4.** Composition of the PLS and O/A ratios used for the validation simulations.

| | REE Concentration (g/L) | | | | | Organic/Aqueous Ratio (O/A) | | | |
| PLS | La | Pr | Nd | Sm | pH | Test 1 | Test 2 | Test 3 | Test 4 |
| --- | --- | --- | --- | --- | --- | --- | --- | --- | --- |
| 1 | 4.86 | 0.55 | 1.67 | 0.088 | 1.7 | 0.7 | 1.5 | 3.0 | 6.0 |
| 2 | 17.7 | 1.97 | 6.12 | 0.327 | 2.3 | 0.5 | 1.0 | 2.0 | 5.0 |
| 3 | 8.82 | 0.98 | 3.08 | 0.164 | 1.4 | 0.5 | 1.0 | 2.0 | 5.0 |

Figure 7 shows the observed and predicted equilibrium pH obtained for the tested O/A ratios from Table 4. The simulated equilibrium pH is calculated with Equation (11). As the pH is an indirect measurement of the quantity of extracted REEs, it can be expected that the predicted extractions will be consistent with the observed ones. This can be verified in Figure 8 for the three PLS of Table 4.

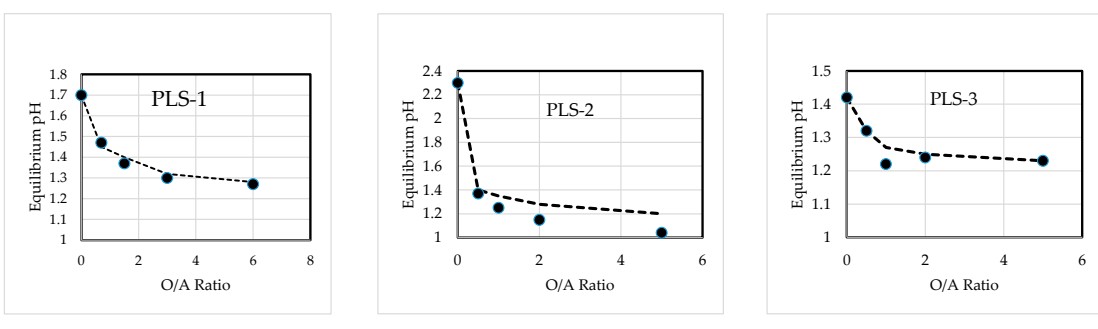

**Figure 7.** Predicted and observed equilibrium pH (symbols = Obs.; Line = Simulated).

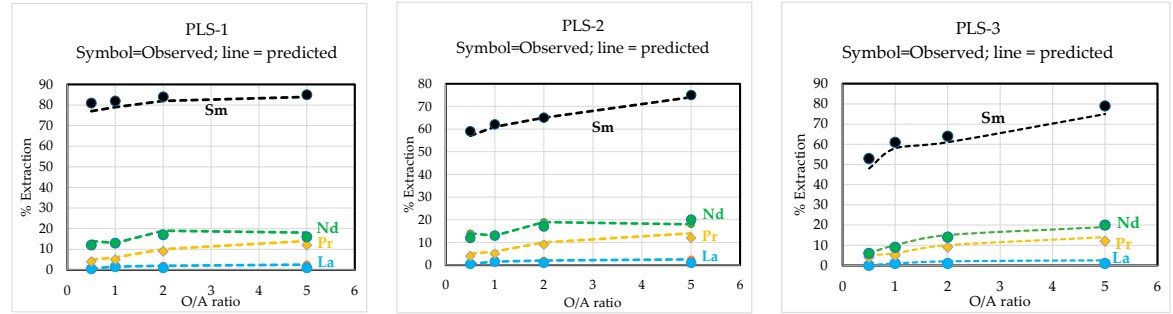

**Figure 8.** Predicted and observed REE extractions (symbols = Obs.; Line = Simulated).

The capability of the model calibrated using extraction test results to reproduce the scrubbing has also been verified [28] confirming that equilibrium constants estimated from batch extraction tests can also be used to simulate the scrubbing of a loaded organic phase. These results for batch tests motivated the extension of the simulation approach to predict the performance of continuously operated circuits as discussed in the following sections.

### 4.2. Validation Using Pilot Plant Data

As a second validation of the simulation methodology, the simulator was extended to predict the operation of a continuous SX circuit. As few organizations around the world have REE-SX pilot plant facilities in operation that can be used to generate the information required to validate the simulator's capability to predict the operation of a continuous circuit, it was not possible to carry out our own pilot plant validation tests, forcing us to use published data. The validation data come from a pilot plant study conducted by SGS [9]. The piloted SX circuit is shown in Figure 9 with the necessary input variables for the simulation. The circuit aims at separating La + Ce from Pr + Nd. The extractant, EHEHPA (commercialized under the name of IonQuest 801, P507, and PC-88A) was initially saponified to 40% and then down to 36%, in order to reduce Ce extraction [9]. The composition of the PLS and circuit operating conditions are given in Figure 9. The equilibrium constants used for the simulation are those reported in Table 1 and estimated from independent batch extraction tests conducted with the same extractant [15]. Once the circuit was deemed stabilized, SGS conducted detailed sampling and chemical analysis of all the aqueous streams of the circuit [9], including the aqueous solution discharged from each settler of the extraction and scrubbing batteries. This information is not used here to calibrate the simulator but to assess its capability to reproduce the observed trends in a continuous REE-SX separation circuit. If a batch extraction process can be readily simulated using an Excel Microsoft™ spreadsheet, the simulation of a continuous process with multiple countercurrent MS required programming the simulation tool. The simulator was coded in Pascal, in the Delphi environment of Embarcadero™, which can accommodate the object-oriented programming well adapted to the simulation approach discussed above.

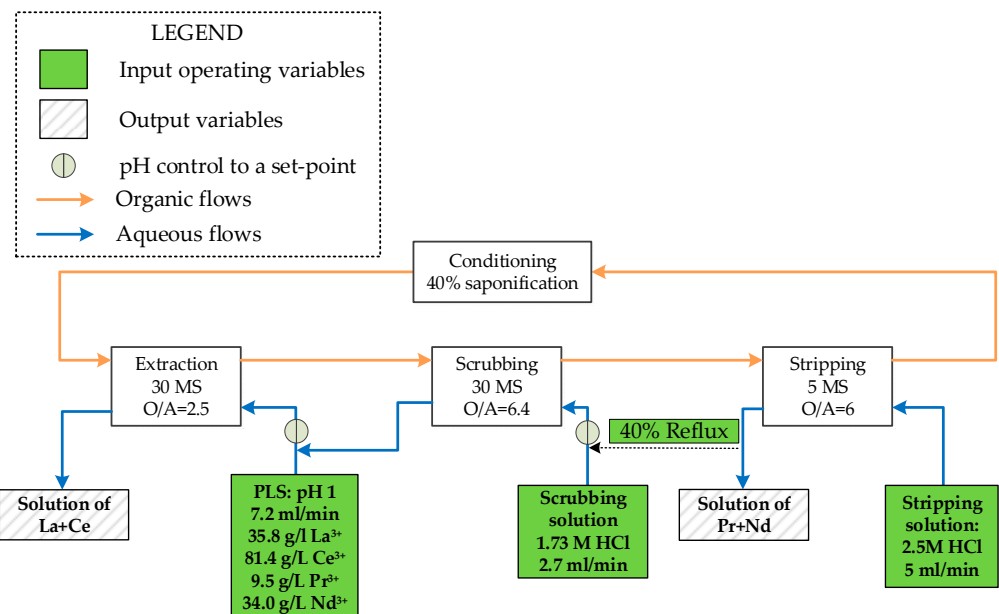

**Figure 9.** Piloted La-Ce/Pr-Nd separation circuit and input variables for the simulation, adapted from [9].

Figure 10 compares the observed and simulated concentrations of La, Ce, Pr, and Nd in the aqueous discharge of each MS of the extraction battery. The PLS enters MS-30 and the extraction raffinate exits MS-1. The discrepancy between the observed and simulated assays at the input of the battery and the composition of the PLS of Figure 10 is due to the reflux from the raffinate scrubbing for which the reflux proportion is not provided and was approximately adjusted for the simulation. Despite the difference in the feed composition of the extraction battery, the simulation reproduces the general trends in the extraction of the La, Ce, Pr, and Nd. The behavior of Ce, which should not be extracted with Pr and Nd, is widely discussed by the authors of the piloting test work [9] and is

attributed to the presence of Ce$^{4+}$. The increase in the Ce concentration in the early portion of the battery is mainly due to changes in the pH along the MS of the battery and the high amount of free extractant available to extract elements in the first MS and little Pr-Nd remaining in the aqueous phase. Experimental results show that the pH varied from 0.8 to 0.5 between MS-3 and MS-30 and then to 1.2 in MS-1 and MS-2 because of saponification. A pH above 1.0 is conducive to Ce extraction along with Pr and Nd, while a pH below 0.8 causes a release of Ce from the loaded organic phase. It should also be mentioned that none of the available REE-SX simulation methods mention the simulation of saponification. Likewise, it should be indicated that the extractant concentration in the organic phase is 50% $v/v$ [9], which is significantly above the 10% concentration used for the extraction tests conducted to calibrate the equilibrium constants [33] and thus may be out of the ideal solution applicability range. It is also possible that the equilibrium conditions are not achieved in the mixers of the pilot plant due in part to partial bypassing, as expected in conditions close to perfect mixing.

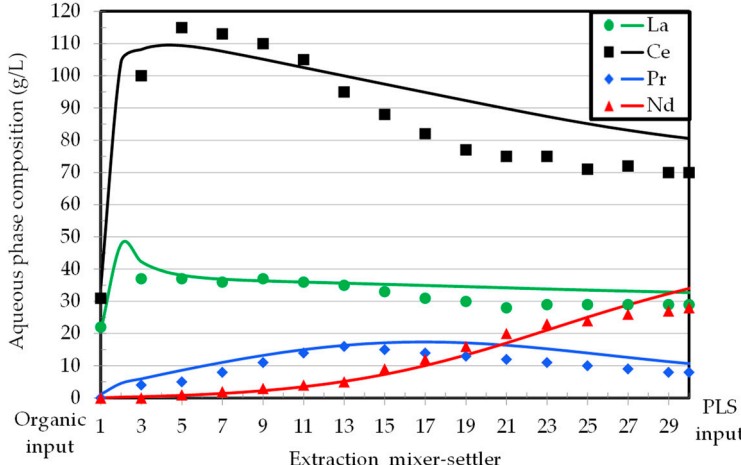

**Figure 10.** Observed and simulated compositions of the aqueous phase at the feed of each MS of the extraction battery (symbols = observed; line = simulated).

The simulation results for the scrubbing battery are shown in Figure 11. The fit between observed and simulated results is good, even though the pH of the scrubbing solution is not explicitly provided by the authors of [9] who indicated having modified the acidity of the scrubbing solution from 3.5 M, to 2.3 M, and to 1.3 M HCl during the 25 days pilot trial (the simulation assumed 1.73 M HCl). The agreement between the observed and simulated results was achieved without the need to adjust the equilibrium constants that were estimated from independent batch extraction tests; a result that still supports the hypothesis of ideal solutions on the basis of the simulation approach. The discrepancy observed for Nd and Pr is due to the reflux from the stripping for which the authors [9] did not provide the actual proportion used during piloting.

The discussion presented in the paper describing the pilot test work [9] identifies numerous difficulties associated with the operation of a REE-SX pilot plant. One of these difficulties is related to the long response time to changes in the operating conditions, which complicates the identification of the optimum pH for scrubbing and the saponification conditions. Moreover, it should be noted that although two campaigns of more than 20 days each were conducted, the test work has not been used to assess the optimum numbers of MS for the extraction or scrubbing batteries, while these numbers are design variables that control the purity of the separated products and drive the capital cost [14]. These experimental difficulties and time-demanding tests confirm the advantages of carrying out a simulation prior to engaging in piloting or full-scale operation. The last section of the paper shows how simulation can indeed reduce the experimental effort by allowing a

rapid screening of the circuit design and operating variables, in order to identify optimum piloting conditions prior to the actual test work.

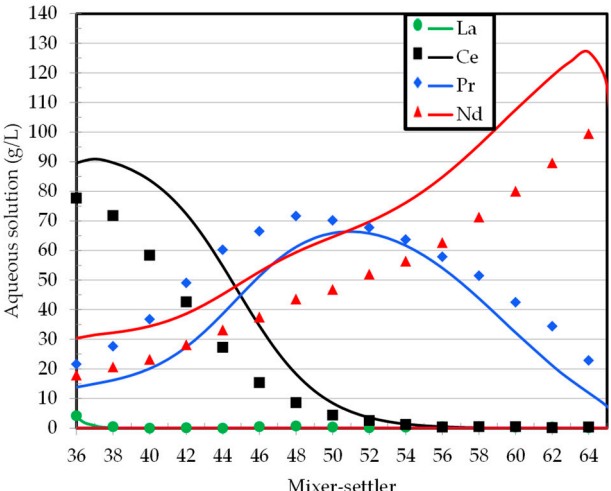

**Figure 11.** Observed and simulated pilot plant aqueous phase composition along the scrubbing battery (the scrubbing aqueous solution enters MS 65 and the loaded organic enters MS-35).

### 4.3. Validation with an Industrial Process

The REE-SX simulator is further validated with the simulation of an industrial REE separation circuit. The difficulty associated with such validation is related to the very limited availability of operating industrial data in the public literature. However, it was possible to obtain some operating data for the pre-2002 Mountain Pass REE separation circuit shown in Figure 12 [1,24] and to reconstruct approximately the composition of the PLS, as given in Table 5. The circuit uses HDEHP (10% *v/v*) as extractant in kerosene. Rather than using a scrubbing stage, the circuit first uses a roughing circuit to separate LREEs from Sm-Lu. The strip solution is then processed to precipitate $Fe^{3+}$ before feeding a cleaning circuit that completes the removal of the LREEs to obtain a strip solution fairly free of LREEs. The cleaning circuit can be viewed as a scrubbing step. The raffinate solution from the cleaning (scrubbing) circuit is circulated back to the roughing circuit to allow the recovery of the Sm-HREEs lost in the cleaning (scrubbing) circuit, similar to the reflux from the scrubbing to the extraction battery. The strip solution of the cleaning circuit contains Sm and heavier REEs and feeds the europium recovery circuit where europium is precipitated out of the solution following a reduction to $Eu^{2+}$ on a zinc mercury column [1,24]. The removal of europium creates a hole between Sm and Gd that facilitates the subsequent SX separation of samarium from Gd and the other HREEs [24]. The circled numbers of Figure 12 correspond to the main streams of the circuit.

The equilibrium constants of P507 were used for simulations and assumed equal to those of HDEHP that were not estimated so far, but this provides the basic information to simulate the whole SX circuit, providing that an estimate of the composition of the solution arriving from the ore preparation circuit can be obtained. Since the detailed PLS composition (all REEs and impurities) is not available in the literature, it was approximately calculated from the composition of bastnaesite of the ore and from the provided europium content of 0.2 g/L of the pregnant leach solution [24]. The composition is obviously approximate for non-Eu REEs, but the proportions of Eu to the non-Eu elements in the solution are consistent with the proportions of these elements in bastnaesite, which is the main mineral carrying REEs of the Mountain Pass ore. Cerium is an exception, as this element is mostly removed prior to the SX circuit [1,24].

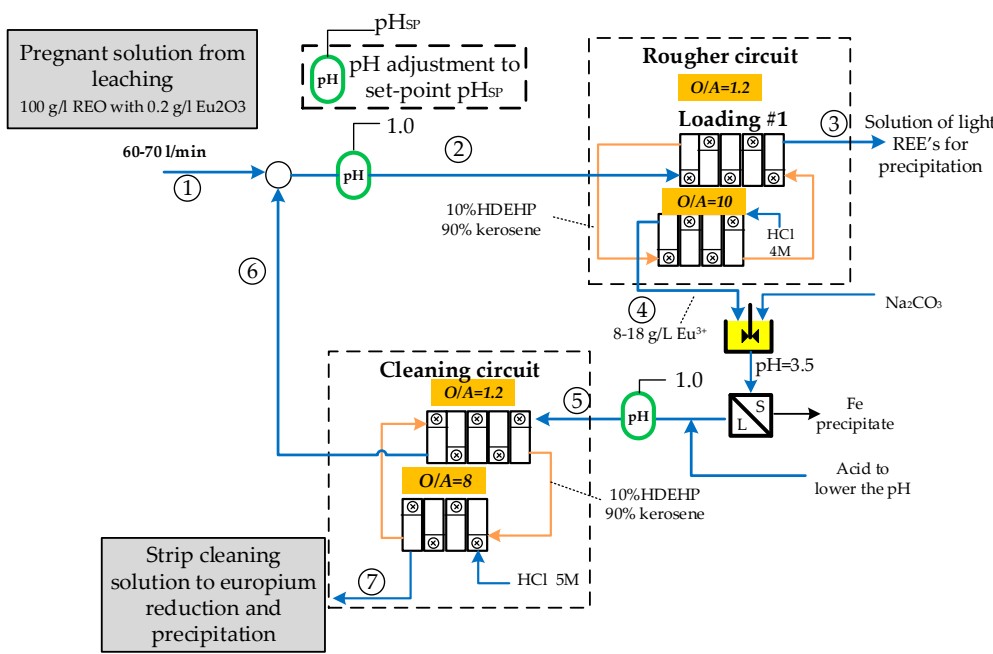

**Figure 12.** Pre-2002 Mountain Pass REE-SX separation circuit, adapted from [1,24].

**Table 5.** Estimated composition of the PLS for the Mountain Pass separation circuit (Cerium is precipitated prior to the SX circuit).

| Concentration (g/L) | | | | | | | | | | | | | | |
|---|---|---|---|---|---|---|---|---|---|---|---|---|---|---|
| Total | $La^{3+}$ | $Ce^{3+}$ | $Pr^{3+}$ | $Nd^{3+}$ | $Sm^{3+}$ | $Eu^{3+}$ | $Gd^{3+}$ | $Tb^{3+}$ | $Ho^{3+}$ | $Er^{3+}$ | $Tm^{3+}$ | $Yb^{3+}$ | $Lu^{3+}$ | $Y^{3+}$ |
| 90.95 | 57.31 | 2.64 | 7.06 | 22.14 | 1.07 | 0.187 | 0.36 | 0.031 | 0.02 | 0.027 | 0.003 | 0.011 | 0.002 | 0.09 |

The objective of the simulation is to verify the consistency of the simulated estimates with reported figures [1,24] such as a 98% Eu recovery in the circuit as well as a 9 to 18 g/L Eu in the solution feeding the iron precipitation circuit (See Figure 12). The simulation conditions (pHs, O/A ratios, and number of MS) are given in Figure 12.

The simulation using the equilibrium constants estimated for an extractant comparable to HDEHP yielded a rougher strip solution (stream 5 in Figure 12) containing 11.5 g/L $Eu^{3+}$, which is in the reported range of 9 to 18 g/L. The $Fe^{3+}$ precipitation is simulated using the solubility constant of $Fe(OH)_3$. REE elements are assumed to remain in solution at the 3.5 $Fe^{3+}$ precipitation pH [1]. The simulated overall europium recovery of the rougher circuit is 98.5%, which compares very well with the reported 98% Eu recovery [1]. However, to obtain the enrichment from 0.2 g $Eu^{3+}$/L in the PLS to more than 10 g $Eu^{3+}$/L in the strip solution, it was necessary to decrease the O/A ratio of the extraction battery and increase the O/A ratio of the rougher circuit stripping battery from 10 to 15. Indeed, the circulating load from the cleaning circuit (stream 6 in Figure 12) assays more than 5 g/L of Eu that still needs to be extracted in the rougher circuit with the Eu from the PLS, explaining the increased demand for extractant in the rougher circuit. It was also necessary to increase the extractant concentration of the organic phase to 20% *v/v*, compared to the reported 10% *v/v* [24]. The simulated compositions of the main circuit streams are given in Table 6. The LREE solution from the rougher circuit is sent for the precipitation and contains less than 1 g/L of HREE, mainly Sm at 0.44 g/L, confirming that the SEG and HREEs are well extracted in the circuit.

**Table 6.** Simulated compositions of the main streams of the Mountain Pass circuit (refer to Figure 12 for the stream #, assays in g/L).

| # | Stream | pH | La$^{3+}$ | Ce$^{3+}$ | Pr$^{3+}$ | Nd$^{3+}$ | Sm$^{3+}$ | Eu$^{3+}$ | Gd$^{3+}$ | Tb$^{3+}$ | Dy$^{3+}$ | Ho$^{3+}$ | Er$^{3+}$ | Tm$^{3+}$ | Yb$^{3+}$ | Lu$^{3+}$ | Y$^{3+}$ | Fe$^{3+}$ |
|---|--------|----|-----|-----|-----|-----|-----|-----|-----|-----|-----|-----|-----|-----|-----|-----|-----|-----|
| 1 | Preg. Leach solution | 1.00 | 57 | 0.005 | 7.6 | 22.4 | 1.07 | 0.19 | 0.4 | 0.031 | 0.049 | 0.020 | 0.027 | 0.003 | 0.011 | 0.002 | 0.090 | 0.002 |
| 2 | Feed to rougher circuit | 1.00 | 53.42 | 0.005 | 7.124 | 21.00 | 2.95 | 0.43 | 0.7 | 0.030 | 0.046 | 0.019 | 0.025 | 0.003 | 0.010 | 0.002 | 0.084 | 0.002 |
| 3 | Raff. From rougher circuit | 1.00 | 49.39 | 0.004 | 6.585 | 19.41 | 0.44 | 0.00 | 0.0 | 0.000 | 0.000 | 0.000 | 0.000 | 0.000 | 0.000 | 0.000 | 0.000 | 0.000 |
| 4 | Strp soln. from rougher | −0.26 | 0.03 | 0.000 | 0.03 | 0.15 | 66.05 | 11.28 | 18.5 | 0.788 | 1.225 | 0.500 | 0.674 | 0.070 | 0.228 | 0.050 | 2.221 | 2.499 |
| 5 | Feed to cleaning circuit | 1.00 | 0.03 | 0.000 | 0.03 | 0.14 | 62.75 | 10.72 | 17.6 | 0.749 | 1.164 | 0.475 | 0.640 | 0.067 | 0.216 | 0.047 | 2.110 | 0.047 |
| 6 | Raff from Cleaning circuit | 0.12 | 0.03 | 0.000 | 0.03 | 0.14 | 49.44 | 6.38 | 9.1 | 0.012 | 0.000 | 0.000 | 0.000 | 0.000 | 0.000 | 0.000 | 0.000 | 0.000 |
| 7 | Soln to Eu ppt. | −0.70 | 0.00 | 0.000 | 0.00 | 0.00 | 3.45 | 1.12 | 2.2 | 0.191 | 0.302 | 0.123 | 0.166 | 0.017 | 0.056 | 0.012 | 0.547 | 0.012 |

The limited amount of available data makes difficult a more rigorous comparison. However, the good prediction of the europium behavior in the circuit is an indication of the robustness of the proposed simulation approach. Moreover, the simulation exercise provides a rapid approach to anticipate, for example, the effect of the circulating load and to assess different corrections such as the adjustment of the O/A in a battery of MS or to the concentration of the extractant in the organic phase.

### 4.4. Validation: Discussion

The three validation examples support the hypothesis that the SX separation of REEs is not an empirical process that needs extensive experimentation to predict the process outcome. Indeed, the various confrontations presented above do not contradict the model working hypotheses that the REE mass transfers between the aqueous and organic phases can be described by equilibrium constants combined with the ideal solution hypothesis. Moreover, even if this last hypothesis posed an issue, the use of activity coefficients could correct and provide additional robustness for highly charged solutions.

## 5. Simulation of the REE-SX Separation Process

This section shows applications of the simulator for planning the piloting of a REE separation circuit and designing by simulation a nine-product REE Separation plant. Currently, the simulator is used to improve the understanding of the operation of REE-SX separation and for assessing ways to reduce the operating and capital costs of a plant.

### 5.1. Application of the Simulator for the Planning of a Pilot Plant Test Work

The simulator can be used to help plan a REE separation piloting study. Running a pilot plant consisting of more than 30 MS is a challenge that few research centers around the world can readily undertake. Indeed, despite 20 days of piloting a LaCe/PrNd separation [9], the impact of the numbers of MS in the extraction and scrubbing has not been studied, leaving also unexplored the effect of the pH of the extraction and scrubbing batteries. Simulation can efficiently be used to rapidly assess design and operating conditions that can subsequently be tested in a pilot plant. As an illustration, the LaCe/PrNd separation previously discussed is considered (Figure 9). The purpose of a separation is to obtain a 99.5% target purity for the two output aqueous streams. A simple but robust purity index for the extraction raffinate that should contain only La and Ce is given by:

$$Z_{La-Ce} = \frac{x_{La}^{XRaf} + x_{Ce}^{XRaf}}{x_{La}^{XRaf} + x_{Ce}^{XRaf} + x_{Pr}^{XRaf} + x_{Nd}^{XRaf}} \tag{17}$$

where $x_e^{XRaf}$ stands for the concentration of element e in the extraction raffinate. Obviously, if the preceding LREE/HREE performed a good separation, then the concentrations of Sm up to Lu should be close to zero and are not included in the denominator of the index. Similarly, a purity index can be defined for the strip solution (*SS*) as:

$$Z_{Pr-Nd} = \frac{x_{Pr}^{SS} + x_{Nd}^{SS}}{x_{La}^{SS} + x_{Ce}^{SS} + x_{Pr}^{SS} + x_{Nd}^{SS}} \tag{18}$$

Ideally, the tuning of a CESS circuit should yield purity indices close to 100% for the two solutions. These purity indices not only measure the quality of the product, but also measure the recovery of the elements in the right product. Indeed, a purity index of 100% for the two aqueous products of the separation indicates a recovery of 100% of the elements in the right product.

The simulator is used here to illustrate an example of pre-screening the operating conditions for the La-Ce/Pr-Nd separation circuit of Figure 9. The composition of the feed solution is given in Figure 9. P507 is the extractant in a 50% $v/v$ concentration in sulfonated kerosene as diluent. The assessed design and operating variables or *factors* are identified in Table 7 with the tested values and the simulated purity indices. For the sake of brevity, only nine conditions (the analysis excluded the O/A ratio in the three batteries, the extractant concentration in the organic phase, the reflux proportions, the number of MS, and the acid concentration for the stripping) are tested here. Four parameters or factors are modified (the pHs of the PLS and the scrubbing feed solution and the numbers of MS units in the extraction battery and the scrubbing battery). Other parameters, such as the extractant concentration and the O/A ratio, were kept constant to simplify the analysis. The raffinate from the scrubbing stage is completely returned and mixed with the PLS in the extraction stage. No reflux from the stripping to the scrubbing is used, while in practice, a significant saving of acid is possible by this reflux that can readily be simulated with the developed simulator.

**Table 7.** Tested simulation conditions and simulated results for the tuning of the La-Ce/Pr-Nd separation circuit.

| Simulated Conditions | Factors | | | | Performance Indices | |
|---|---|---|---|---|---|---|
| | Number of Extraction MS | Number of Scrubbing MS | pH in the Extraction | pH in the Scrubbing | La + Ce Purity | Pr + Nd Purity |
| 1 | 10 | 10 | 1.0 | 0.0 | 99.9 | 46.6 |
| 2 | 30 | 10 | 1.0 | −0.5 | 89.6 | 100.0 |
| 3 | 10 | 30 | 1.0 | −0.5 | 59.5 | 100.0 |
| 4 | 30 | 30 | 1.0 | 0.0 | 100.0 | 46.7 |
| 5 | 10 | 10 | 2.0 | −0.5 | 94.3 | 99.9 |
| 6 | 30 | 10 | 2.0 | 0.0 | 100.0 | 43.5 |
| 7 | 10 | 30 | 2.0 | 0.0 | 10.0 | 43.5 |
| 8 | 30 | 30 | 2.0 | −0.5 | 94.3 | 100.0 |
| 9 | 20 | 20 | 1.0 | −0.3 | 100.0 | 65.3 |

The nine separations conditions of Table 7 were simulated in less than 30 min. Considering that it takes 14 days to stabilize a pilot plant for one condition, more than 3 months would have been required to pilot the nine conditions. Simulation results indicate that acceptable purities can be achieved with a combination of 10 MS for the extraction battery and 10 MS for the scrubbing battery (minimize CapEx) and using an extraction pH of 2.0 and a scrubbing pH of −0.5 (3N). However, the 94.3% purity for the La-Ce product indicates losses of Pr and Nd to the raffinate. Piloting this condition could then be used to validate the simulation and confirm the feasibility of the proposed flowsheet and conditions. Finally, the decision will be made based on the economics of the process. Considering that lanthanum and cerium 2021 prices [40] of 1.37 USD/kg $La_2O_3$ and 1.41 USD/kg $CeO_2$ are marginal compared to those of praseodymium and neodymium (137 USD/kg $Pr_6O_{11}$; 145 USD/kg $Nd_2O_3$), it may prove uneconomic to achieve a 100% purity in the PrNd product. The addition of an economic index to the simulator is progressing, so that the simulation will not only provide the composition of the REE solutions, but the benefits generated by a given operation of a separation circuit. Some illustrations of the application of such an economic index are presented in [28].

## 5.2. Simulation of a Complete Separation Plant

As indicated above, most of the simulation studies presented in the literature are limited to one separation (ex. LREE/HREE). The simulation tool presented here can be used to design a complete separation circuit and this application is illustrated in this section. Table 8 gives the composition of the separation plant feed solution after digestion in HCl of 10 t/day of oxide concentrate obtained by processing a Canadian REE ore [28]. The PLS concentrations in Table 8 sum up to 150 g/L, which is typical for a REE-SX Plant [41]. The separation circuit is designed to process 37.3 L/min of PLS fed at pH 2. The maximum value contained in the solution is 9.50 USD/L or 354 USD/min or 50,976 USD/t of REE oxide, assuming a perfect separation of all the elements into their individual oxides and prices given in Table 8. Based on some NI-43-101 for REEs [17,18], refining charges for such concentrate could be as high as 35% of the contained value or refining charges of USD 18,000 per tonne of oxide concentrate. The simulator can be used to estimate the operating and capital costs associated with the operation of a plant to help in taking the go/no-go decision with the construction and operation of a separation plant.

**Table 8.** Composition of the REE solution for the simulated separation plant with REE oxide prices [40].

| | La | Ce | Pr | Nd | Sm | Eu | Gd | Tb |
|---|---|---|---|---|---|---|---|---|
| Concentration in the solution (g/L) | 27.65 | 56.89 | 7.1 | 27.16 | 3.89 | 0.16 | 3.56 | 0.48 |
| | Dy | Ho | Er | Tm | Yb | Lu | Y | |
| | 3.08 | 0.65 | 1.63 | 0.16 | 1.15 | 0.16 | 16.29 | |
| | La | Ce | Pr | Nd | Sm | Eu | Gd | Tb |
| 2021 Price (USD/kg oxide) | 1.37 | 1.49 | 137.46 | 144.92 | 4.56 | 31.42 | 71.87 | 1759.43 |
| | Dy | Ho | Er | Tm | Yb | Lu | Y | |
| | 454 | 205.8 | 53.8 | 126.46 | 20 | 832.6 | 12.02 | |

The nine-product separation circuit is shown in Figure 13. The number of MS for the extraction, scrubbing, and stripping batteries of each separation circuit (SX in Figure 13) and the operating conditions (pH, ratio O/A, etc.) are adjusted to have a 99.5+% purity (see Equations (17) and (18)) for each final separated REE. The extractant used in all the separation circuits is P507 in a 10% *v/v* solution in kerosene. The equilibrium constants for the extractant are given in Table 1. A bulk product of HREE (Er to Lu) is produced for selling to a custom refinery or it could be stockpiled awaiting a favorable market for the separated elements [27]. The bulk HREE (Er to Lu) product is not considered as a payable product. The holmium oxide is also not considered as a payable product and was thus not targeted for separation at a high purity. Nd and Pr are not separated as there is a market for bulk Pr-Nd oxide [16]. The simulation and tuning (selection of the number of MS and operating conditions) of the unitary separation circuits (SX- in Figure 13) are performed sequentially and manually, but there is a real potential in undertaking a global circuit design and tuning supervised by an optimization algorithm using an economic index [28]. Table 9 summarizes the final tuning and designed conditions for the nine-product separation circuit. For the sake of brevity, only the main parameters are given in Table 9. The proportions of reflux are not indicated, but the strategic use of reflux for saving acid in the scrubbing will be studied in detail in a subsequent paper. The pessimistic estimates for the consumption of acid (HCl 30% *w/w*) and caustic (NaOH 50% *w/w*) are also given in Table 9. Indeed, substantial acid savings are possible by the strategic use of reflux from the stripping to the scrubbing batteries. The simulation also shows that significant acid savings in the stripping batteries can be achieved by an early removal of HREEs [28]. Caustic is mainly used for saponification and pH control.

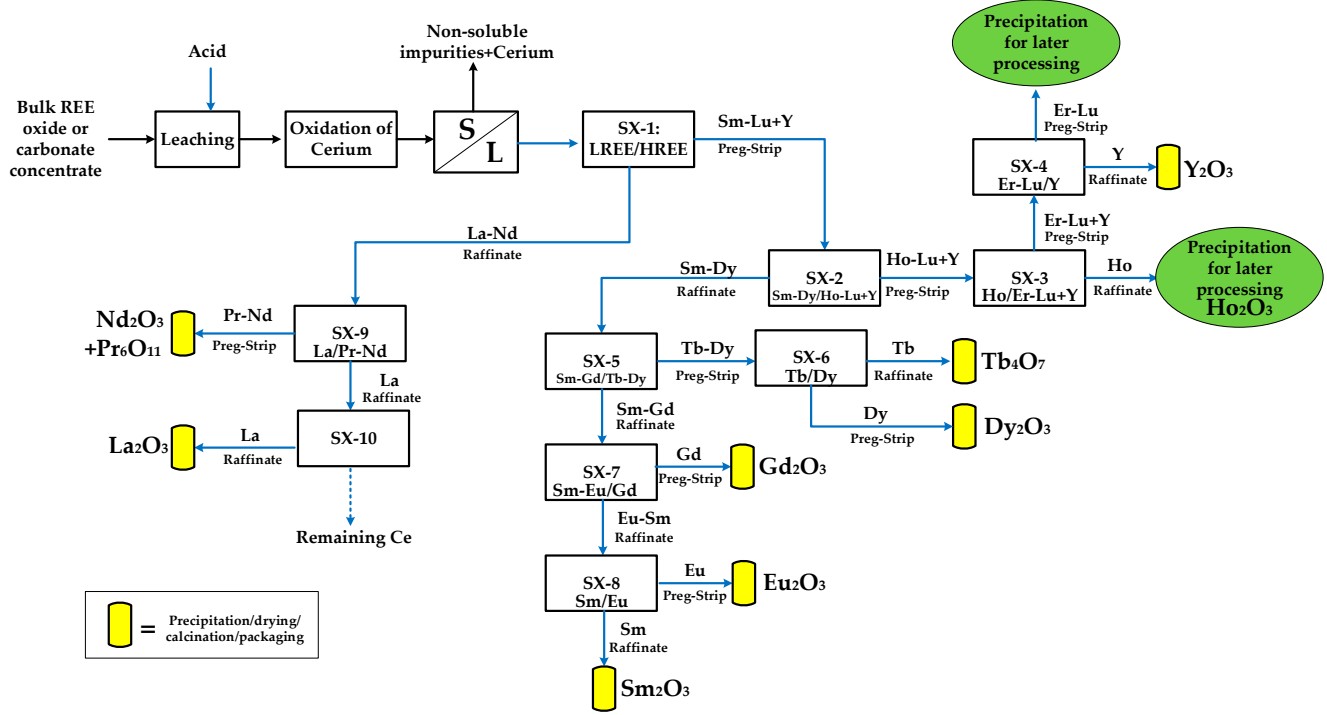

**Figure 13.** Nine-Product separation circuit.

**Table 9.** Design and operating conditions for the 9-product separation circuit.

| Circuit | | Number of MS | | | | O/A Ratio | | | Reagent Consumption (t/t Oxide REE Conc) | |
|---|---|---|---|---|---|---|---|---|---|---|
| | Separation | Ext | Scrb | Strp | Tot. | Ext | Scrb | Strp | HCl (30% *w/w*) | NaOH (50% *w/w*) |
| SX-1 | LREE/HREE | 14 | 10 | 17 | 41 | 2 | 7.6 | 2.5 | 6.70 | 0.57 |
| SX-2 | Sm-Dy/Ho-Lu-Y | 20 | 20 | 17 | 57 | 1.4 | 7.6 | 2.5 | 7.13 | 2.23 |
| SX-3 | Ho/Er-Lu-Y | 15 | 15 | 17 | 47 | 1.4 | 10 | 2.5 | 7.16 | 2.77 |
| SX-4 | Er-Lu/Y | 30 | 30 | 17 | 77 | 1 | 5 | 2.5 | 8.52 | 4.37 |
| SX-5 | Sm-Gd/Tb-Dy | 10 | 10 | 20 | 40 | 1.1 | 10 | 5 | 2.56 | 0.98 |
| SX-6 | Tb/Dy | 16 | 8 | 20 | 44 | 2.7 | 10 | 5 | 2.98 | 0.79 |
| SX-7 | Sm-Eu/Gd | 30 | 30 | 10 | 70 | 1 | 20 | 20 | 0.85 | 0.89 |
| SX-8 | Sm/Eu | 8 | 6 | 4 | 18 | 0.9 | 20 | 20 | 0.70 | 0.33 |
| SX-9 | La-Ce/Pr-Nd | 30 | 40 | 4 | 74 | 2.75 | 9.6 | 20 | 3.77 | 1.85 |
| SX-10 | La/Ce | 24 | 40 | 4 | 68 | 3 | 7.9 | 25 | 4.10 | 2.54 |
| Total | | 197 | 209 | 130 | 536 | | | | 44.47 | 17.32 |

Results of Table 9 show that a total of 536 MS are required for the separations. This already provides the necessary information to calculate a first estimate of the capital costs for the project. The number of MS for the stripping batteries of all SX circuits is large to ensure a complete stripping of the heaviest REEs (Er to Lu) from the organic phase. Indeed, the simulation shows that if the heavy REEs are not stripped completely from the organic phase, they accumulate in the organic phase and reduce its extraction capacity [42]. The stripping of these heavy REEs also requires a large proportion (75%) of the acid used in the circuit, justifying investigations into approaches to remove the heaviest REEs in the early stage of a complete separation circuit in order to reduce the capital and operating costs of a plant [28].

Table 10 gives the daily revenues generated by the products, calculated using the oxide prices of Table 8. The most revenue comes from the bulk Pr-Nd oxides followed by the dysprosium and terbium oxides (shown in bold in Table 10). The lanthanum and

cerium oxides generate marginal revenues despite their separation requiring 68 MS or 13% of the total MS of the circuit with acid and caustic consumptions of, respectively, 10% and 15% of the plant requirements (Table 9). The La/Ce separation circuit is clearly a loss-making operation that yearns for an early removal of cerium [30]. The data of Table 9 also show that the separation of the SEG (Sm/Eu/Gd) into individual oxides requires 88 MS and is responsible for 9% and 10% of the acid and caustic consumed by the plant. The plant capital and operating costs can likely be reduced by precipitating Eu following a reduction to $Eu^{2+}$ as practiced in some plants [1,24]. The precipitation of Eu out of the SEG solution creates a «hole» between Sm and Gd that facilitates and significantly reduces the cost of the separation of Sm and Gd. Indeed, preliminary simulations show that this separation could be achieved in less than 20 MS, instead of 88, when Eu is not precipitated. Unfortunately, there is very little information on the equipment and costs associated with the operation of Eu reduction and precipitation, making it difficult to conduct a detailed economic assessment of this operation.

**Table 10.** Revenues generated by the products of the 9-product separation plant.

|  | $La_2O_3$ | $CeO_2$ | $Pr_6O_{11} +$ $Nd_2O_3$ | $Sm_2O_3$ | $Eu_2O_3$ | $Gd_2O_3$ | $Tb_4O_7$ | $Dy_2O_3$ | $Y_2O_3$ | Total |
|---|---|---|---|---|---|---|---|---|---|---|
| Throughput kg/day | 1754 | 3768 | 2138 | 238 | 10 | 221 | 30 | 191 | 1109 | 9459 |
| Revenues USD/day | 2402 | 5614 | **306,527** | 1086 | 310 | 15,863 | **52,032** | **86,667** | 13,306 | 483,808 |

Once the separation plant flowsheet is established, the simulator can be used to study the impact of the feedstock composition on the revenues generated by a custom REE-SX separation plant and to assess processing options to maximize the operating benefits according to the element concentrations in the feed.

## 6. Conclusions

Unlike mineral processing circuits whose operation and design are more an art than a science, the separation of REEs lends itself well to a fundamental description of the mass transfer reactions governing the process. This paper shows that the separation of REEs by solvent extraction can be adequately modelled and simulated by relying on the fundamental description of the process using a limited number of working hypotheses and by keeping the experimental effort to a minimum. The use of simulation for a REE-SX pre-feasibility study can significantly reduce the effort for piloting and help promoters to assess processing options for a REE ore deposit or can simply improve the understanding of the operation of a REE-SX separation.

**Author Contributions:** Conceptualization, K.T. and C.B.; methodology, K.T. and J.-F.B.; software, K.T.; validation, K.T. and C.B.; formal analysis, J.-F.B.; investigation, K.T. and C.B.; data curation, J.-F.B.; writing—original draft preparation, C.B.; writing—review and editing, K.T., J.-F.B. and C.B.; visualization, J.-F.B.; supervision, C.B; project administration, C.B.; funding acquisition, C.B. All authors have read and agreed to the published version of the manuscript.

**Funding:** This research was funded by Quebec FRQNT 2014-MI-181215 and the Canada NSERC/CRD program and Canmet of Natural Resources Canada.

**Data Availability Statement:** The data presented in this study are available on request from the corresponding author.

**Acknowledgments:** The authors acknowledge the technical assistance of Vicky Dodier and the collaboration of Dominic Larivière and Laurence Whitty-Léveillé for the fine-tuning of the methodology for assaying rare earth elements. The authors are also grateful to John R. Goode for his kind advice concerning the operation of REE-SX plants.

**Conflicts of Interest:** The authors declare no conflict of interest. The funders had no role in the design of the study; in the collection, analyses, or interpretation of data; in the writing of the manuscript; or in the decision to publish the results.

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
