# Peer review of "Simulation of Solvent Extraction Circuits for the Separation of Rare Earth Elements"

_minerals, doi:10.3390/min13060714_

Round 1
Reviewer 1 Report
This paper describes a simulation method requiring a minimum of calibration effort and which can be used to design a complete REE separation plant. The work is novel and well presented.
Author Response
Point 1: This paper describes a simulation method requiring a minimum of calibration effort and which can be used to design a complete REE separation plant. The work is novel and well presented.
Response 1: Little modifications following comment.
Reviewer 2 Report
Manuscript minerals-2388781 covers important topic of REE separation using SX technique. Authors proposed mathematical model based on equilibrium constants, which may be used for simulation of SX process. It is a very valuable publication which may be interesting to readers interested in this field. There are only some minor remarks which may be addressed prior to publication:
- Line 57 “piloting a circuit”?
- Lines 138-139 – in my opinion “E” should be capital letter instead of “X” for explanation of CapEx and OpEx abbreviations
English used throught the text is OK.
Author Response
Point 1: Line 57 “piloting a circuit”?
Response 1: added a definition of piloting a circuit, (small-scale implementation of the process).
Point 2: Lines 138-139 – in my opinion “E” should be capital letter instead of “X” for explanation of CapEx and OpEx abbreviations.
Response 2: modified the text accordingly to the suggestion.
Reviewer 3 Report
Overall it is an interesting paper on the industrial separation of rare earth elements. The introduction is a bit much for a journal article, but, on another hand, it will serve as a reference material for future work. The simulation calculations are well presented, but experimental details are lacking (see below).
a few light edits:
line 156: "SX" has already been defined.
the authors use the term "pregnant solutions". This reviewer is not familiar with such term. Is there another word that would suit?
table 4: define "PLS"
lines 522, 528: who are the "authors" mentioned?
Other comments:
While the presentation of the problem is rather lengthy, there is a lack of information on the experimental portion of the work. There is a lack of description of the experimental materials and equipment. Figure 8 shows pH around 1. Please describe the electrode used. Was it a glass electrode. How was the electrode calibrated? If it was calibrated using the typical colored buffer (4, 7, 10), the calibration does not cover pH 1. Furthermore, glass electrodes are notoriously imprecise for pH that low and will render erroneous simulations.
Please provide the vendor for the chemical used. Was de-ionized water used to prepare the solutions? How was the AES calibrated?
How were the extraction conducted: were the sample shaken? if yes, how were they shaken and for how long? How were pH adjusted to 1.7, 2.3, 1.4 (table 4)-I suspect these are initial pH?
Table 4 shows several "cases" with different volume ratios. I do not see how this translate in the results.
Author Response
Point 1: Overall it is an interesting paper on the industrial separation of rare earth elements. The introduction is a bit much for a journal article, but, on another hand, it will serve as a reference material for future work. The simulation calculations are well presented, but experimental details are lacking (see below).
Response 1: Indeed, the introduction is lengthy, but as you mentioned, it will serve as a reference for future work. Added a methodology section to further describe the details of the experimental tests.
Point 2: line 156: "SX" has already been defined.
Response 2: Removed the redundant definition of SX.
Point 3: The authors use the term "pregnant solutions". This reviewer is not familiar with such term. Is there another word that would suit?
Response 3: Replaced “pregnant solution” by the more rigorous term of “Pregnant Leach Solution (PLS)”, i.e. a solution from the leaching of a concentrate. Also labelled the figure 2 with the term PLS for the corresponding flow (the solution entering the SX circuit).
Point 4: table 4: define "PLS"
Response 4: added a definition of PLS, Pregnant Leach Solution (PLS).
Point 5: lines 522, 528: who are the "authors" mentioned?
Response 5: added clarification, authors of the reference article [9]
Point 6: While the presentation of the problem is rather lengthy, there is a lack of information on the experimental portion of the work. There is a lack of description of the experimental materials and equipment. Figure 8 shows pH around 1. Please describe the electrode used. Was it a glass electrode. How was the electrode calibrated? If it was calibrated using the typical colored buffer (4, 7, 10), the calibration does not cover pH 1. Furthermore, glass electrodes are notoriously imprecise for pH that low and will render erroneous simulations.
Response 6: Added a methodology section to further describe the details of the experimental tests. The pH of the aqueous solutions is measured using an accumet XL600 pH-meter with an Orion ROSS pH probe calibrated from pH 1 to 7 (Thermo Fisher scientific).
Point 7: Please provide the vendor for the chemical used. Was de-ionized water used to prepare the solutions? How was the AES calibrated?
Response 7: added the vendor for chemicals, water type used (de-ionized) and AES calibration method in the methodology section.
Point 8: How were the extraction conducted: were the sample shaken? if yes, how were they shaken and for how long? How were pH adjusted to 1.7, 2.3, 1.4 (table 4)-I suspect these are initial pH?
Response 8: added how the extractions were conducted. For the extraction tests, about 50 ml of aqueous solution is mixed with the organic phase at a specific organic to aqueous volume ratio in separatory funnel of 250 ml, then agitated mechanically for 10 min on a VWR rocking shaker. Also added how pH were adjusted (with HCl and indeed, these are initial pH).
Point 9: Table 4 shows several "cases" with different volume ratios. I do not see how this translate in the results.
Response 9: to clarify that point, replaced “case” by “test” and referenced the Table 4 in the next paragraph to link the observed and predicted results presented in Figure 7 and Figure 8 to the test conditions described in Table 4.